# Tumor Microenvironment before and after Chemoradiation in Locally Advanced Rectal Cancer: Beyond PD-L1

**DOI:** 10.3390/cancers15010276

**Published:** 2022-12-31

**Authors:** Pritam Tayshetye, Andrew J. Friday, Ashten N. Omstead, Tanvi Verma, Stacey Miller, Ping Zheng, Prashant Jani, Ali Zaidi, Gene Finley

**Affiliations:** 1Department of Hematology-Oncology, Allegheny Health Network, Pittsburgh, PA 15212, USA; 2Department of Medical Oncology, Allegheny General Hospital, Pittsburgh, PA 15212, USA; 3Esophageal and Lung Institute, Allegheny Health Network, Pittsburgh, PA 15212, USA; 4Department of Pathology and Laboratory Medicine, Allegheny Health Network, Pittsburgh, PA 15212, USA; 5Hematology and Oncology, Northeast Cancer Centre, Sudbury, ON P3E 5J1, Canada

**Keywords:** rectal cancer, chemoradiation, tumor microenvironment, neoadjuvant, biomarker, immunotherapy

## Abstract

**Simple Summary:**

Localized rectal cancer is currently managed with neoadjuvant therapy before surgery which includes concurrent chemoradiation therapy alone or a total neoadjuvant therapy approach which involves chemotherapy and concurrent chemoradiation sequentially. These strategies are changing, and optimal management of rectal cancer continues to evolve. Immunotherapy has entered the treatment paradigm in advanced rectal cancer for patients with microsatellite instability. A role for immunotherapy for early-stage disease has yet to be established. Clinical trials in rectal cancer incorporating immunotherapy in the neoadjuvant settings are underway however better understanding of the tumor microenvironment and targeting specific biomarkers may be more efficacious. In this paper, we report on rectal cancer tumor microenvironment changes following neoadjuvant chemoradiation. Identifying changes in biomarker expression within the tumor microenvironment may be predictive of better outcomes, improved response to immunotherapy and may also identify new targets which could lead to targeted therapeutic drug development.

**Abstract:**

Background: In locally advanced rectal cancer treatment, neoadjuvant concurrent chemoradiation therapy (cCRT) is the standard of care. The tumor microenvironment (TME) is a complex entity comprising of tumor cells, immune cells and surrounding stroma and is closely associated with tumor growth and survival, response to antitumor therapies and also resistance to treatment. We aimed to assess the change in biomarkers associated with TME following standard neoadjuvant cCRT in rectal cancer. Methods: We accessed archival tissue from rectal cancer patients treated with neoadjuvant cCRT at Allegheny Health Network (AHN) facilities over the past 14 years. Pre-treatment and post-treatment biopsies were assayed for PD-L1, CD8+ T-cells, CXCL9, TIM-3, IDO-1, IFN-G, IL17RE, LAG-3, and OX40 in 41 patients. Results: We found statistically significant upregulation in multiple biomarkers namely CD8, IL17RE, LAG3 and OX40 post neoadjuvant cCRT and a trend towards upregulation, although not statistically significant, in biomarkers PD-L1, CXCL9, TIM-3, IDO-1 and IFN-G expression. Conclusions: This provides a glimpse into the TME before and after neoadjuvant cCRT. We suggest that the biomarkers noted to be upregulated could be used for designing appropriate clinical trials and development of therapeutic targeted drug therapy in an effort to achieve better response to neoadjuvant therapy, increasing clinical and pathological complete response rates and improved overall outcomes.

## 1. Introduction

Colorectal cancer (CRC) is the third most common cancer in both women and men. It is estimated that 151,030 new cases of CRC will be diagnosed in 2022. It is also estimated that 52,580 people will die from CRC in 2022 [1]. In total, 41% of rectal cancers diagnosed from 2009–2015 were localized and 34% were regional, with a 5-year survival rate of 89% and 71%, respectively [2].

The current standard of care in non-metastatic rectal cancer involves neoadjuvant concurrent chemoradiation therapy (cCRT), utilizing capecitabine or infusional 5-fluorouracil with radiation, followed by surgery and subsequently adjuvant chemotherapy. More recently, a total neoadjuvant therapy (TNT) approach utilizing both chemotherapy and cCRT in a neoadjuvant manner followed by surgery is also frequently utilized for eligible patients with locally advanced disease [3]. This approach is also an acceptable option as per National Comprehensive Cancer Network guidelines for treatment of locally advanced rectal cancer [4]. Surgery is generally performed 6–8 weeks after the completion of cCRT. Studies have also investigated extending duration between cCRT and surgery beyond 6–8 weeks may lead to an increase in pathologic complete response (pCR) without increasing the risk of surgical complications [5]. 

Evidence linking the pathological response to neoadjuvant treatment and long-term outcomes has been established multiple times in the literature [6,7]. One meta-analysis of over 3105 patients with rectal cancer across 14 data sets evaluated correlation between pCR rate and long-term outcomes. Five-year disease-free survival was 83.3% for patients with pCR and 65.6% for those without pCR [8]. Patients that do not achieve pCR with standard treatment may have tumors with more aggressive biology that warrant an altered regimen [9]. 

We believe that the tumor microenvironment (TME) has an important role to play in response to treatment and even in achieving pCR. It is well known that TME is complex and involves interactions between tumor cells, immune cells and surrounding stroma [10]. At our institution, we previously tested the hypothesis that cCRT upregulates PD-L1 expression in rectal cancer. Additionally, there was an associated increase in the number of CD8+ T-cells in the biopsies after treatment [11]. We believe that improving the pCR rates in these patients may be achievable with the combination of novel immunotherapy agents and radiotherapy in the neoadjuvant setting eventually leading to longer overall survival of these patients. Clinical trials are currently ongoing which are exploring this hypothesis.

In this study, we aimed to assess for change in the other biomarkers of TME namely CXCL9, TIM-3, IDO-1, IFN-G, IL17RE, LAG-3, and OX40 in addition to PD-L1 and CD8+ T-cells. We hypothesize that if other biomarkers of TME show upregulation, it may pave way for future targeted drug development and clinical trial design in an effort to achieve better response to neoadjuvant therapy, increasing pCR rates and improved overall outcomes.

## 2. Materials and Methods

### 2.1. Study Design

Each patient who received neoadjuvant cCRT, followed by surgical resection (n = 41) consented to having tissue biopsies and resected tissue samples used for the study. Additionally, two patients without CRC consented to having their normal colonic mucosa samples used as control tissues for gene expression analyses. Each sample, biopsy and resection tissue was formalin-fixed paraffin-embedded (FFPE). 

### 2.2. Laser Microdissection and Gene Expression

We followed the procedures previously described by Matsui et al. (2016) for the collection, preparation, and genetic analysis of tumorous tissues [12]. FFPE tissue sections were cut 5 μm thick using a microtome, and sections were placed on irradiated polyethylene napthalate (PEN) membrane slides. Each slide was stained with cresyl violet and was subjected to laser microdissection within 30 min of staining. Tumorous tissues identified by a board-certified pathologist were collected from each PEN membrane slide using a Leica LMD6500 laser microdissection system (Leica, Wetzlar, Germany). Cut tissues were placed in 150 μL of deparaffinization solution and stored at −80 °C until gene expression assays were conducted. 

Prior to RT-PCR, RNA containing miRNA was isolated and quantified using the miRNeasy FFPE Kit (Qiagen, Valencia, CA, USA; #217504). A minimum of 20 μg of RNA was then reverse transcribed using the miScript II RT kit (Qiagen, Valencia, CA, USA; #218161). The cDNA products were preamplified with the miScript PreAmp PCR kit (Qiagen, Valencia, CA, USA; #331452), diluted 20-fold, and then stored at −20 °C until subjected to RT-PCR. We used the miScript SYBR Green PCR Kit (Qiagen, Valencia, CA, USA; #218075) to perform RT-PCR on each sample, using the following seven RT^2^ primer assays: LAG3 (Qiagen, Valencia, CA, USA; #PPH01035A), TIM-3 (Qiagen, Valencia, CA, USA; #PPH00583A), IDO1 (Qiagen, Valencia, CA, USA; #PPH01328B), IFN-G (Qiagen, Valencia, CA, USA; #PPH00380), OX40 (Qiagen, Valencia, CA, USA; #PPH00818A), IL17RE (Qiagen, Valencia, CA, USA; #PPH18962A), and CXCL9 (Qiagen, Valencia, CA, USA; #PPH00700B). We included two housekeeping genes, SNORD95 and miR-16, to serve as endogenous controls and included normal colonic mucosa on each plate to serve as the quality control. Each sample was performed in a technical duplicate and was normalized against the normal tissue and the control housekeeping genes. 

### 2.3. Statistical Analyses

Categorical variable PD-L1 was presented in frequency (n) and percentage (%), while eight continuous variables, CD8+ T-cells, CXCL9, TIM-3, IDO-1, IFN-G, IL17RE, LAG-3, and OX40 were presented in number (N), mean and standard deviation [Mean (SD)], median and interquartile range [Median (IQR)]. A normality test, e.g., Shapiro–Wilk test, was performed for each of the eight biomarkers, respectively. Log transformation was applied to the data if the normality test for the data was failed (*p* < 0.05).

Univariate analyses were performed for estimating changes in the biomarkers. Specifically, a generalized estimating equations model was employed to estimate the difference in proportion for post-treatment PD-L1 staining compared to pre-treatment, while eight mixed models were applied to estimate changes in values for the other eight biomarkers, respectively. Statistical significance was set at *p* < 0.05.

All statistical analyses were conducted in SAS software (version 9.4; SAS Institute, Cary, NC, USA).

## 3. Results

Except for PD-L1 staining, the data for the eight biomarkers was very sparse and failed the normality test (*p* < 0.05). Therefore, log transformation was applied, which resulted in normally distributed data. Summary of all the nine biomarkers is presented in Table 1. 

One patient had two post-treatment samples at different time points, so total sample number for post treatment included in the analyses is n = 42. For PD-L1 and CD8 staining, 2 post-treatment cases were omitted due to inadequate tissue or tumor available on the block. 

For RT-PCR testing, a few cases were omitted due to inadequate tissue or a low RNA yield from the dissected tissue, which prevented us from proceeding with RT-PCR analysis resulting in 38 pretreatment samples and 37 post treatment samples. 

The univariate analyses revealed statistically significant increase in multiple biomarkers namely CD8, IL17RE, LAG3 and OX40 post neoadjuvant cCRT. There were borderline significant changes in CXCL9, TIM-3, IDO-1 and IFN-G expression and a trend toward upregulation of these genes was noted which was however, not statistically significant. Additionally, PD-L1 staining was 2.2 times higher post-treatment compared to the pre-treatment value but not statistically significant (Table 2).

## 4. Discussion

TME as stated previously consists mainly of tumor cells, tumor stromal cells, immune cells and non-cellular components of extracellular matrix [13]. All components of TME interact with each other during tumorigenesis and also with effects of therapy. Research over the last decade has demonstrated that radiotherapy, which is a critical part of rectal cancer treatment, plays an important role in immunomodulation via several mechanisms [14]. Firstly, the destruction of tumor cells by radiation increases tumor immunogenicity by causing the release of molecules known as Tumor Associated Antigens (TAAs) as cells are broken down. This effect is complimented by radiation-induced upregulation of cell surface receptors for TAA molecules, thus allowing for TAA recognition by Antigen Presenting Cells (APCs), such as dendritic cells and macrophages. These APCs endocytose the TAAs, process them, and present them to tumor-specific cytotoxic T-cells via major histocompatibility complex-1 (MHC-1) proteins, thereby priming the immune system against tumor evasion and proliferation [15].

Tumor destruction caused by radiotherapy leads to the release of cytokines that upregulate co-stimulatory signaling to naive T-cells. Moreover, radiation induces cellular adhesion molecules on tumor vascular endothelium, promoting recruitment, chemotaxis and infiltration of cytotoxic T-cells into the TME [15]. All of these processes together can strengthen the immune response against tumor proliferation and spread. 

During tumorigenesis, proliferating tumor cells must evade antitumor immunity for survival [16]. Tumors can develop resistance to the immune system response by “co-opting” certain immune-checkpoint pathways, especially those involved in T-cell and TAA interaction, by reducing expression of TAAs and promoting anti-apoptotic signaling [17]. There exist subpopulations of tumor cells that can undergo genomic changes and altered neo-antigen presentation, which helps them evade and survive antitumor activity [18]. 

To accomplish this, tumor cells rely in part on the activation of negative regulatory pathways, such as the PD-1/PD-L1 checkpoint. This interaction has been well documented between tumor cells and host T-cells. When the surface receptor, PD-1, on T-cells comes into contact with its ligand, PD-L1, being expressed in the TME; this interaction transmits an inhibitory signal through interactions with SHP-1/2 phosphatases [16]. Although PD-L1 expression was upregulated but not statistically significant in our analysis, it appears that as tumor cells rely more and more on PD-1/PD-L1 interaction to evade immune surveillance; these tumor cells become more sensitized, and thus more susceptible, to PD-1/PD-L1 blockade. Currently, the clinical trials cooperative group, National Surgical Adjuvant Breast and Bowel Project (NSABP), has completed a phase II trial (FR-2) assessing the activity of PD-L1 inhibition with durvalumab (MEDI4736) following cCRT and prior to surgery in patients with Stage II-IV microsatellite stable rectal cancer and results are awaited. This trial will potentially reinforce the concept that patients with tumors susceptible to PD-1/PD-L1 blockade can benefit from the additive effects of immunotherapy following cCRT prior to surgery.

In our study, a statistically significant increase in CD8+ T-cells, IL-17RE, LAG-3 and OX40, following cCRT was noted, whereas there was a trend toward upregulation of other genes, which included PD-L1, CXCL9, TIM-3, IDO-1 and IFN-G expression, but it was not statistically significant. These upregulated genes regulate signaling pathways that can influence tumor cell fate decisions within the TME and as such could prove to be prime targets for site-directed immunotherapy inhibition paired with neoadjuvant cCRT to promote better patient remission and quality of life.

As a component of the immune system, CD8+ T-cells can kill tumor cells with cytotoxic molecules, such as granzymes and perforin. IFNγ, which is produced by CD8+ T-cells, can increase the expression of MHC class I antigens by tumor cells, thereby rendering them better targets for CD8+ T-cells. In a meta-analysis of 23 studies on tumor infiltrating lymphocytes (TILs), Gooden et al. observed that the presence of CD8+ TILs resulted in a prognostic advantage for all survival endpoints tested [19]. Antitumor activity of CD8+ TILs and their favorable effect on disease recurrence and survival has been demonstrated earlier [20]. Galon et al., in 2006, concluded that the immunological data (the type, density, and location of immune cells within the tumor samples) was found to be a better predictor of patient survival than the histopathological methods used to stage CRC [21]. The correlation between high frequency of CD8+ TILs and the lack of lymph node involvement in a cohort of early rectal cancer was noted by Daster et al. [22]. Our study demonstrated significant upregulation in CD8+ TILs in the TME post-cCRT. The antitumor activity of the upregulated CD8+ TILs could be potentiated further with therapeutic modulation of other components of the TME.

Interleukin 17 Receptor-E (IL-17RE) is a cytokine produced by Th17 cells, a T-helper cell subset developed from an activated CD4+ T-cell. Main role of IL-17 in humans is in host pathogen defense, in particular to extracellular bacterial and fungal infections [23]. In regard to cancer, tumorigenesis is promoted by IL-17 via the combination of releasing myeloid-derived suppressor cells (MDSCs) to inhibit immune system, as well as stimulating pro-inflammatory cytokines to maintain an inflammatory environment. This results in the stimulation of tumor growth via the subsequent expression of anti-apoptotic genes and the consequent increased survival of cells with pro-tumorigenic potential. IL-17 was also shown to enhance metastasis via the expression of vascular endothelial growth factor inducing both angiogenesis and lymphangiogenesis, subsequently leading to metastasis of tumors [24]. Our analysis reports significant upregulation of IL-17RE after cCRT which could potentially be explored as a therapeutic target. Inhibition of IL-17 could theoretically inhibit tumorigenesis and curb metastasis. 

OX40 has been shown to regulate tumor immunity as a member of the tumor necrosis factor superfamily and is highly expressed by activated CD4, CD8 T-cells and regulatory T-cells (Tregs) and to a lesser extent by neutrophils and natural killer (NK) cells. OX40 stimulates activating T-cells and enhances their ability to proliferate and survive along with modulation of NK cell function. OX40 stimulatory signals are also sufficient to reverse T-cell tolerance by inhibiting the suppressive activity of Tregs, promoting survival and expansion of T-cells. Currently monoclonal antibodies are being developed which activate OX40 to potentially increase the antitumor activity of immune cells within the TME. OX40 expression was significantly upregulated after cCRT in our study, which suggests a role for OX40 activating antibodies to be used post cCRT in an effort to achieve better disease control prior to surgical resection [25,26,27].

Lymphocyte activating gene 3 (LAG-3) is an immune checkpoint receptor expressed in both activated cytotoxic T-cells and Tregs as well as NK cells. T-cell MHC class II molecules downregulate T-cell proliferation following LAG-3 binding. LAG-3 is often co-expressed with PD-1 and it has a negative regulatory effect over T-cell function, preventing tissue damage and autoimmunity. Cancer cells utilize this function to escape immune-mediated destruction which in turn leads to tumor growth. In our analysis, LAG-3 was noted to be upregulated after cCRT which can potentially be targeted by LAG-3 inhibitors. Thus LAG-3 blockade is being investigated along with PD-1 blockade as a therapeutic target in ongoing clinical trials [27,28].

CXCL9 is a member of the chemokine CXC family that has a role in the chemotaxis of immune cells. However, role of CXCL9 remains unclear and contradictory. It can have both, tumor-promoting and tumor-suppressive roles even in the same cancer type [29]. In CRC, specifically, prior studies have shown a relatively low expression of CXCL9 [30]. However, CXCL9 has been correlated with T-cell infiltration and prolonged disease-free survival in CRC [31]. In our analysis, CXCL9 expression increased post cCRT but was not statistically significant. However, utilizing CXCL9 mediated T-cell infiltration with other immune checkpoint inhibitors can be explored. 

IDO-1 (indoleamine 2,3-dioxygenase) is a rate-limiting enzyme in the tryptophan catabolism pathway. IDO-1’s role in immunosuppression has been well documented during gestation where it protects the fetus from the maternal active immune system [32]. Similarly in the TME, IDO-1 promotes the activation and recruitment of MDSCs and Tregs [33,34]. Additionally, IDO-1 has been implicated in tumor vascularization via IFN-G. IDO-1 and IFN-G appear to have co-stimulatory roles, promoting the induction/expression of each other in tumor immunoevasion [35]. IFN-G, however, has been shown to have seemingly conflicting roles in modulating both tumor immunoevasion and suppression of tumor growth. IDO-1 has increased expression in multiple tumor cell types and is relatively highly expressed in colorectal tumors [36]. Micro-RNA downregulation of IDO-1 has shown an inverse relationship between IDO-1 and patient survival. Some promising trials have shown that IDO-1 immunotherapy works in conjunction with cCRT to enhance the effectiveness of treatment [37,38]. IFN-G inhibition, however, has shown mixed results in patients with the same tumor type [39]. This can likely be attributed to the duality of IFN-G’s role in pro and antitumor immunosuppression [40]. Although not significantly upregulated in our analysis post cCRT, the functions of IDO-1 and IFN-G make them tempting targets for continued immunotherapy studies paired with cCRT.

TIM-3 (T-cell immunoglobulin-3) is expressed on NK cells and macrophages and promotes immunosuppression by inhibiting them and also by promoting expansion of MDSCs. Elevated levels of TIM-3+ T-cells have been shown to negatively affect prognosis in non-small cell lung cancer and follicular lymphoma [27,41]. The kinetics of TIM-3 upregulation during PD-1 blocking treatment is thought to potentially represent a resistance mechanism leading to treatment failure with PD-L1/PD-1 antibodies. In our patient population, TIM-3 expression was upregulated but was not statistically significant. Radiation may help to make tumor cells more homogeneous with expression of these proteins which could be exploited [42].

More recently, the therapeutic efficacy of immunotherapy in harnessing the TME was demonstrated in 12 patients with mismatch repair deficient locally advanced rectal cancer who were treated with dostarlimab, a PD-1 inhibitor, and all patients achieved a 100% complete clinical response [43]. This study reinforces the potential of TME and use of immunomodulatory therapies to achieve disease control.

## 5. Conclusions

As noted in our study, there is variable change in multiple biomarkers of the TME in rectal cancer after neoadjuvant cCRT. Some of these biomarkers are immunosuppressive or have pro-tumor activity such as PD-L1, IL17RE, LAG-3, IDO-1, TIM-3 whereas others have antitumor activity such as CD8+ T-cells, OX40, CXCL9 and others have a mixed role such as IFN-G.

Our lab’s main drive is to combine the positive effects of both cCRT and biomarker driven immunotherapy in patients diagnosed with rectal cancer. We propose that these patients could benefit from the combined effects of neoadjuvant cCRT followed by targeted biomarker driven immunotherapy by suppressing the ability of tumorous cancer cells to evade immune-mediated killing. We propose a strategy in which immunotherapy will consist of biomarker driven therapy to inhibit the tumor cells’ ability to evade immune surveillance and detection which subsequently will lead to tumor cell destruction. With the efforts of cCRT and site-directed immunotherapy working synergistically, we propose that this combined approach will result in sufficient tumor suppression and improved complete clinical response rates and also improved pCR rates that, following surgery, should translate to better patient outcomes, including sustained cancer remission and reduced rates of relapse. Furthermore, we plan to perform additional studies utilizing nanostring molecular diagnostics which will allow for more in-depth analysis of key regulatory genes in cancerous tumor cells pre- and post-cCRT.

## Figures and Tables

**Table 1 cancers-15-00276-t001:** Summary for rectal cancer biomarkers by time point.

**Biomarker**	**Pre-Treatment (n = 40) [n, (%)]**			**Post-Treatment (n = 42) [n, (%)]**		
**PD-L1**						
<1%	36 (90.0)			32 (76.2)		
1–49%	4 (10.0)			8 (19)		
**Biomarker**	**Pre-Treatment (n = 40)**			**Post-Treatment (n = 42)**		
	**N**	**Mean (SD)**	**Median (IQR)**	**N**	**Mean (SD)**	**Median (IQR)**
CD8	40	3.9 (0.8)	3.9 (3.3, 4.4)	42	4.5 (0.8)	4.4 (4.1, 5.2)
CXCL9	38	1.1 (2.4)	0.5 (−0.3, 2.8)	37	2.1 (2.4)	2.2 (1.0, 3.3)
TIM-3	38	0.4 (2.7)	0.4 (−1.6, 2.3)	37	1.6 (2.7)	1.5 (0.1, 3.8)
IDO1	38	2.2 (4.2)	2.6 (−0.2, 5.2)	37	4.1 (4.0)	4.7 (1.9, 7.0)
IFN-G	38	0.9 (3.4)	1.0 (−1.2, 2.6)	37	2.5 (3.4)	3.5 (0.1, 4.9)
IL17RE	38	0.5 (2.6)	0.5 (−1.3, 2.7)	37	2.0 (2.4)	2.4 (0.9, 3.8)
LAG3	38	0.9 (2.9)	1.0 (−1.1, 2.7)	37	2.5 (2.8)	2.9 (0.5, 4.8)
OX40	38	0.5 (2.1)	0.8 (−1.1, 2.0)	37	1.8 (2.2)	2.1 (0.5, 3.4)

**Table 2 cancers-15-00276-t002:** Univariate analyses for change in biomarkers.

Biomarker	Pre-Treatment	Post-Treatment	Estimated Difference (SE)	Estimated Odds Ratio (95% CI) *	*p* Value
	Estimated Mean (SE)	Estimated Mean (SE)			
PD-L1 (ref = Pre-treatment)	−2.2 (0.5) **	−1.4 (0.4) ***	0.8 (0.5)	2.2 (0.7, 6.4)	0.1546
CD8	3.9 (0.1)	4.5 (0.1)	0.6 (0.2)	N/A	**0.0008**
CXCL9	1.1 (0.4)	2.1 (0.4)	1.0 (0.6)	N/A	0.0752
TIM-3	0.4 (0.4)	1.6 (0.4)	1.2 (0.6)	N/A	0.0620
IDO1	2.2 (0.7)	4.1 (0.7)	1.9 (0.9)	N/A	0.0573
IFN-G	0.9 (0.6)	2.5 (0.6)	1.6 (0.8)	N/A	0.0543
IL17RE	0.5 (0.4)	2.0 (0.4)	1.5 (0.6)	N/A	**0.0118**
LAG3	0.9 (0.5)	2.4 (0.5)	1.5 (0.6)	N/A	**0.0259**
OX40	0.5 (0.3)	1.8 (0.3)	1.3 (0.5)	N/A	**0.0100**

* Estimated Odds Ratio (95%CI) was only applied to outcome “PD-L1” where generalized estimating equation was performed, and not applicable (N/A) for other outcomes where mixed models were used. ** Pre-treatment: Based on Estimated Mean (SE) = −2.2 (0.5) estimated probability for the outcome variable “PD-L1” in “1 to 49%” was 10.9% (95% CI: 3.8%, 31.0%). *** Post-treatment: Based on Estimated Mean (SE) = −1.4 (0.4) estimated probability for the outcome variable “PD-L1” in “1 to 49%” was 23.8% (95% CI: 10.5%, 54.1%); *p* Values in bold are statistically significant.

## Data Availability

All data generated or analyzed during this study are included in this published article.

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
