# Peer review of "Tumor Microenvironment before and after Chemoradiation in Locally Advanced Rectal Cancer: Beyond PD-L1"

_cancers, 2022, doi:10.3390/cancers15010276_

Round 1

Reviewer 1 Report

The work by Tayshetye et al reports the identification of potential biomarkers upon chemoradiation therapy. They have successfully concluded that, other than PD-L1, several other biomarkers, such as CD8, IL17RE, LAG3, and OX40 are also upregulated, though not statistically significant. Given the increasingly heavy burden of colorectal cancer, this work can be a critical contribution toward understanding the colorectal tumor microenvironment and may facilitate the future design of drugs by taking advantage of other targets. I, therefore, recommend the publication of this manuscript in Cancers after minor updates.

1) The authors are encouraged to correctly define CRT. There are several definitions in the first two pages. In addition, it would be helpful to define what is CRT as chemotherapy and radiation therapy are two independent processes.

2) Tumor microenvironment is widely abbreviated as TME. These terms including that in Point 1, are repeatedly defined which is unnecessary.

3) Refine the sentence in lines 53-54 (followed by ….followed by…).

Author Response

Response to Reviewer 1 comments

Point 1) The authors are encouraged to correctly define CRT. There are several definitions in the first two pages. In addition, it would be helpful to define what is CRT as chemotherapy and radiation therapy are two independent processes.

Response 1) The reviewer’s comments are greatly appreciated. CRT in our study refers to concurrent/combined use of chemotherapy and radiation. We have replaced the CRT term with cCRT to stand for concurrent chemoradiation therapy to make it easier for readers to understand and to avoid any confusion. When only chemotherapy is mentioned, it refers to the use of chemotherapy without combining radiation.

2) Tumor microenvironment is widely abbreviated as TME. These terms including that in Point 1, are repeatedly defined which is unnecessary.

Response 2) We appreciate the reviewer’s recommendation and we have therefore changed the abbreviation of tumor microenvironment from TM to TME throughout the manuscript. Since page 1 has the abstract and the full article text starts from page 2 we chose to define cCRT (concurrent chemoradiation therapy) and TME (tumor microenvironment) terms both in abstract and in the introduction of the full text for ease of the readers as some readers may choose to read only the abstract whereas some may read the full article starting from the introduction.

3) Refine the sentence in lines 53-54 (followed by ….followed by…).

Response 3) The sentence is now modified from “The current standard of care in non-metastatic rectal cancer involves neoadjuvant chemoradiation therapy (CRT), using capecitabine or infusional 5-fluorouracil concurrent with radiation followed by surgery followed by adjuvant chemotherapy” to the following:

“The current standard of care in non-metastatic rectal cancer involves neoadjuvant concurrent chemoradiation therapy (cCRT), utilizing capecitabine or infusional 5-fluorouracil with radiation, followed by surgery and subsequently adjuvant chemotherapy.”

Reviewer 2 Report

This is an interesting study and the authors are to be commended for their efforts.

Can the authors report on whether there's a relationship between baseline and post-treatment tumor microenvironment biomarkers and clinical outcomes (ie. pathologic and clinical response at minimum; recurrence-free and overall survival if there has been sufficient follow up).  It would be interesting to see these biomarkers have any predictive clinical potential. 

Author Response

Response to Reviewer 2 comments

Point 1) Can the authors report on whether there's a relationship between baseline and post-treatment tumor microenvironment biomarkers and clinical outcomes (ie. pathologic and clinical response at minimum; recurrence-free and overall survival if there has been sufficient follow up).  It would be interesting to see these biomarkers have any predictive clinical potential. 

Response 1) We appreciate the reviewer’s feedback and their comments. This study was performed as a proof of concept study using archival tumor tissue to solely assess change in biomarkers before and after concurrent chemoradiation. As such clinical outcomes was not the primary focus of the study. Also since archival tissue was analyzed, patient charts and follow-up details of 13 patients were not available. However, of the patients that we had the details on, we found that 27 patients (66%) had achieved a complete pathologic response. 7 patients (17%) had recurrent disease. To make a predictive analysis of multiple biomarkers with low number of patient charts having long-term follow-up as such was not feasible. We, however, do agree that it would be interesting to see if the biomarkers had any predictive value in terms of response or survival. We are in fact currently conducting another study, performing a pre and post treatment biomarker analysis of tumor microenvironment of rectal cancer patients with Nanostring technology, utilizing a larger cohort of patients in which clinical variables will be assessed too.